# Exact simulation of pigment-protein complexes unveils vibronic renormalization of electronic parameters in ultrafast spectroscopy

F. Caycedo-Soler[1], A. Mattioni [1], J. Lim[1], T. Renger [2], S. F. Huelga [1]✉ & M. B. Plenio [1]✉

The primary steps of photosynthesis rely on the generation, transport, and trapping of excitons in pigment-protein complexes (PPCs). Generically, PPCs possess highly structured vibrational spectra, combining many discrete intra-pigment modes and a quasi-continuous of protein modes, with vibrational and electronic couplings of comparable strength. The intricacy of the resulting vibronic dynamics poses significant challenges in establishing a quantitative connection between spectroscopic data and underlying microscopic models. Here we show how to address this challenge using numerically exact simulation methods by considering two model systems, namely the water-soluble chlorophyll-binding protein of cauliflower and the special pair of bacterial reaction centers. We demonstrate that the inclusion of the full multi-mode vibronic dynamics in numerical calculations of linear spectra leads to systematic and quantitatively significant corrections to electronic parameter estimation. These multi-mode vibronic effects are shown to be relevant in the longstanding discussion regarding the origin of long-lived oscillations in multidimensional nonlinear spectra.

[1] Institute of Theoretical Physics and IQST, Ulm University, Albert-Einstein-Allee 11, 89081 Ulm, Germany. [2] Institute of Theoretical Physics, Department of Theoretical Biophysics, Johannes Kepler University Linz, Altenberger Str. 69, 4040 Linz, Austria. ✉email: susana.huelga@uni-ulm.de; martin.plenio@uni-ulm.de

ight-harvesting (LH) antennas and photo-chemical reaction
centers (RC) provide the elementary building blocks of the
photosynthetic apparatus of plants, algae, and bacteria[1].
Primarily these molecular aggregates consist of absorbing mole-
cules (pigments) complexed with specific proteins to form a PPC.
Despite its fundamental importance to biology, the dynamical
characterization of these complexes to a degree that can repro-
duce all reported spectroscopic data in a single microscopic
model remains an outstanding challenge.

Reduced models of excitonic dynamics subject to purely thermal
fluctuations can achieve reasonable agreement with linear optical
spectra[2–9]. The quantitative explanation of all relevant aspects of
multi-dimensional nonlinear spectroscopy though requires a more
detailed model of the system-environment interaction that takes into
account the full complexity of the environmental structure[10]. Indeed,
spectroscopic studies of PPCs at low temperatures[11–14] reveal the
presence of vibrational environments that consist of a broad spec-
trum of low-frequency protein modes with room temperature
energy scales, and several tens of discrete high-frequency modes that
originate mainly from intra-pigment dynamics[11,12,15]. Nonlinear
optical experiments on monomer pigments in solution at both
77 K[16,17] and room temperature[18,19], as well as first-principles
calculations[20,21] further corroborate the underdamped nature of
intra-pigment vibrational modes with picosecond lifetimes.

Recently, a range of vibronic models in which pigments are
subject to the combined influence of a broad unstructured
bosonic environment and a small number of vibrational modes
with frequencies in the vicinity of excitonic transitions have been
formulated[22–31]. In this picture, vibrational lifetime borrowing
can lead to long-lasting oscillatory dynamics of coherences
between excitonic states, and observations of long-lasting oscil-
latory features in multi-dimensional spectroscopy[32–37] have been
attributed to this effect[38–43]. Notwithstanding, the identification
of a universally accepted origin of these long-lived oscillations
remains a subject of active discussion[34,44–47].

An important obstacle that prevents the conclusive resolution of
this debate is the fact that the interpretation of spectroscopic data
and their underpinning dynamical features can be influenced sig-
nificantly by the specific choice of electronic and vibrational para-
meters that enter the PPC models. We will demonstrate that by
accounting for the full environmental spectral density, involving
more than 50 intra-pigment modes per site in addition to a broad
background, the presence of high-frequency long-lived vibrational
modes can lead to quantitatively significant modification of the
calculated linear spectra of PPCs and consequently the estimated
values of electronic parameters to recover a best fit with actual
measurements. These corrections do not appear when considering
only selected resonant modes and go well beyond predictions
obtained by using conventional line shape theory[48–51].

To present our results, we provide an analytical theory of
renormalization effects due to multi-mode vibronic mixing in
model excitonic systems of two prototypical PPCs, namely the
water-soluble chlorophyll-binding protein (WSCP) of cauliflower
and the special pair (SP) of bacterial reaction centers, depicted in
Fig. 1. By considering realistic environmental spectral densities,
we corroborate our predictions using two independent numeri-
cally exact methods (the temperature-dependent time evolving
density matrix using orthogonal polynomials algorithm,
T-TEDOPA[22,52–54], and the hierarchical equations of motion,
HEOM[55]). We show that the hybridization of electronic and
vibrational degrees of freedom requires a significant renormali-
zation of electronic couplings. Importantly, this renormalization
of electronic parameters, in turn, is shown to have a significant
impact on the dynamics of excitonic coherences, notably the
lifetimes of their oscillatory dynamics.

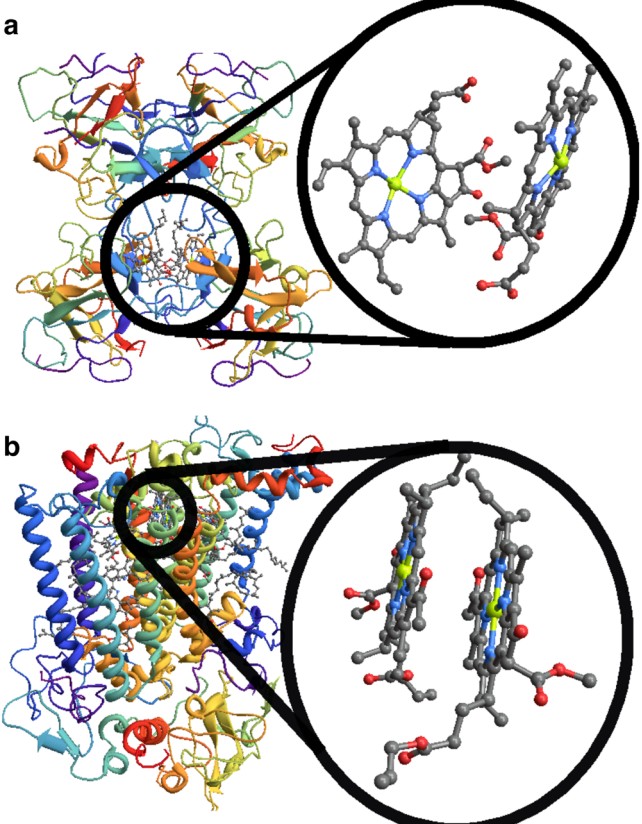

**Fig. 1 Photosynthetic pigment-protein complexes. a** Molecular structure
of water-soluble chlorophyll-binding protein from cauliflower, a natural
dimeric PPC, with Chl*b* homodimer shown in detail. **b** Molecular structure
of bacterial reaction center from purple bacterium *Rb. Sphaeroides* with a
(hetero)-dimeric unit of special pair highlighted. Site energies and couplings
for the relevant pigments are obtained from models that combine the
crystal structure together with a comparison of calculated and measured
spectra[70].

## Results

**Electronic and vibronic couplings of PPCs**. Absorption spectra
of PPCs are determined by the electronic energy-level structure of
pigments, their mutual electronic interactions and the coupling of
the resulting excitons to vibrational degrees of freedom of the
pigment's environment. In the following, we will restrict our
analysis to the $Q_y$ transition between electronic ground and first
excited states of the pigments, which suffices for the evaluation of
the low-energy part of absorption spectra and is relevant for
photosynthetic energy transfer[1]. For the dimeric WSCP and SP,
the electronic Hamiltonian is then described by (see Supple-
mentary Note 1)

$$H_e = \sum_{i=1}^{2} \varepsilon_i |\varepsilon_i\rangle\langle\varepsilon_i| + V(|\varepsilon_1\rangle\langle\varepsilon_2| + |\varepsilon_2\rangle\langle\varepsilon_1|). \quad (1)$$

Here $|\varepsilon_i\rangle$ denotes the singly excited state of site $i$ with on-site
energy $\varepsilon_i$ that is in the visible (WSCP) or in the near infrared
spectrum (SP). The on-site energies depend on their local
environment and therefore suffer from static disorder inducing
ensemble dephasing that will be included in our numerical
treatment. The electronic coupling $V$ leads to delocalized elec-
tronic eigenstates (excitons), $H_e|E_\pm\rangle = E_\pm|E_\pm\rangle$, and an exci-
tonic splitting $\Delta = E_+ - E_- = \sqrt{4V^2 + (\varepsilon_1 - \varepsilon_2)^2}$. In WSCP, the
mean site energies are identical, $\langle\varepsilon_1\rangle = \langle\varepsilon_2\rangle$, due to the symmetry

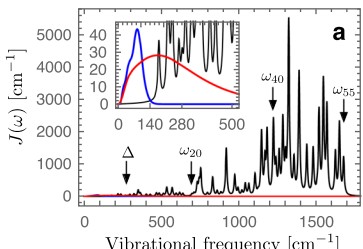 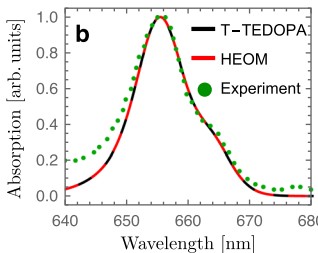 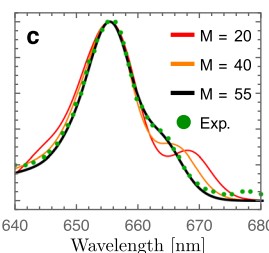 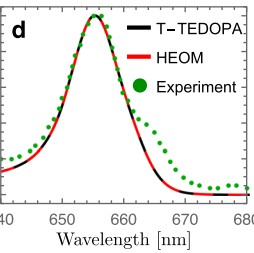

**Fig. 2 Absorption spectra of WSCP. a** Experimentally estimated spectral density of WSCP, consisting of 55 intra-pigment modes $J_h(\omega)$[13] and low-frequency protein modes $J_l^{\mathrm{WSCP}}(\omega)$[58], shown in black and blue, respectively. Experimentally estimated spectral density $J_l^{\mathrm{B777}}(\omega)$ of B777 complexes is shown in red[60]. The position of the excitonic splitting $\Delta = 280\ \mathrm{cm}^{-1}$ obtained for an electronic coupling $V = 140\ \mathrm{cm}^{-1}$ is indicated by a black arrow. The 20th, 40th and 55th lowest vibrational frequencies of the intra-pigment modes are marked by black arrows with $\omega_{20}$, $\omega_{40}$, and $\omega_{55}$, respectively. **b** Experimental absorption spectrum of WSCP at 77 K, shown in green dots, and numerical results obtained by T-TEDOPA and HEOM, shown in black solid and red dashed lines, respectively, for $V = 69\ \mathrm{cm}^{-1}$ and $J_l^{\mathrm{B777}}(\omega)$[60]. **c** For $V = 140\ \mathrm{cm}^{-1}$ and $J_l^{\mathrm{WSCP}}(\omega) + J_h(\omega)$, T-TEDOPA and HEOM results can reproduce the experimental absorption spectrum, as shown in black. Numerically exact absorption spectra for the $M \in \{20, 40, 55\}$ lowest frequency intra-pigment modes are displayed where $M = 55$ represents the full experimentally estimated spectral density. **d** For $V = 69\ \mathrm{cm}^{-1}$ and $J_l^{\mathrm{WSCP}}(\omega) + J_h(\omega)$, T-TEDOPA and HEOM results cannot reproduce the experimental absorption spectra. See Supplementary Note 5 for details of the other molecular parameters used in these simulations. We note that the maximum amplitudes of simulated absorption spectra at 656 nm are normalized to unity for a comparison with experimental absorption line shape.

of molecular structure, while in SP, the mean site energies are different as pigments are surrounded by nonidentical local protein environments. Another difference concerns the electronic coupling strength, which is stronger in SP due to electron exchange giving rise to short-range Dexter type contributions[56,57].

The exciton dynamics of PPCs is driven by vibrational modes that induce fluctuations in the transition energies $\varepsilon_i$ of pigments. The full electronic-vibrational interaction, induced by $N$ vibrational modes per site, is described by the Hamiltonian $H = H_e + H_\nu + H_{e-\nu}$ where

$$H_\nu = \sum_{i=1}^{2} \sum_{k=1}^{N} \omega_k b_{i,k}^\dagger b_{i,k}, \qquad (2)$$

$$H_{e-\nu} = \sum_{i=1}^{2} |\varepsilon_i\rangle\langle\varepsilon_i| \sum_{k=1}^{N} \omega_k \sqrt{s_k}(b_{i,k} + b_{i,k}^\dagger). \qquad (3)$$

Here the annihilation (creation) operator $b_{i,k}$ ($b_{i,k}^\dagger$) describes a local vibrational mode of frequency $\omega_k$ coupled to site $i$ with a strength quantified by the Huang-Rhys (HR) factor $s_k$. For an environment initially in a thermal state, the ensuing dynamics is fully determined by the environmental spectral density $J(\omega) = \sum_k \omega_k^2 s_k \delta(\omega - \omega_k)$ whose structure needs to be determined experimentally or theoretically.

**Structure of the environmental spectral density.** Generally, in PPCs the spectral density $J(\omega)$ consists of a broad background and multiple sharp peaks distributed across a broad range of frequencies. These can be determined by fluorescence line-narrowing (FLN) and hole burning experiments which reveal that the environmental spectral densities of WSCP and SP consist of low-frequency broad features originating from protein motions, and 55 intra-pigment modes resulting in multiple narrow peaks in the high-frequency part of the spectrum. The contribution of the protein modes of WSCP may be described by log-normal distribution functions of the form $J_l^{\mathrm{WSCP}}(\omega) = \sum_m (\omega c_m/\sigma_m) \exp(-[\ln(\omega/\Omega_m)]^2/2\sigma_m^2)$, which provides a satisfactory description of the low-energy part of experimentally measured FLN spectra of WSCP[58]. Alternatively, the protein motions of WSCP have been modeled by the following functional form: $J_l^{\mathrm{B777}}(\omega) = \frac{S}{s_1+s_2}\sum_{i=1}^{2}\frac{s_i}{7!2\omega_i^4}\omega^5 e^{-(\omega/\omega_i)^{1/2}}$ that has been extracted from FLN spectra of B777 photosynthetic complexes[59] and considered in the simulations of WSCP[60]. Every underdamped intra-pigment mode contributes a

Lorentzian of width $\gamma_k \sim 1\ \mathrm{ps}^{-1}$, resulting in $J(\omega) = J_l(\omega) + J_h(\omega)$ where

$$J_h(\omega) = \sum_{k=1}^{55} \frac{4\omega_k s_k \gamma_k (\omega_k^2 + \gamma_k^2)\omega}{\pi((\omega + \omega_k)^2 + \gamma_k^2)((\omega - \omega_k)^2 + \gamma_k^2)}, \qquad (4)$$

and the reorganization energy of the high-frequency modes is given by $\lambda_h = \int_0^\infty d\omega J_h(\omega)/\omega = \sum_{k=1}^{55} \omega_k s_k$. The reorganization energy of the 55 intra-pigment modes of WSCP[13] (SP[15]) is 660 cm$^{-1}$ (379 cm$^{-1}$), which is several times larger than that of quasi-continuous protein spectrum[58,61] and quasi-resonant intra-pigment modes with $\omega_k \approx \Delta$ (see Supplementary Note 5). The presence of underdamped vibrational modes can lead to long-lived correlations between electronic and vibrational degrees of freedom that make the rigorous numerical treatment of the ensuing vibronic dynamics very costly. In non-perturbative HEOM simulations, where experimentally or theoretically estimated spectral densities are fitted by the sum of Drude–Lorentz peaks[21,62], the simulation cost of a dimeric system exceeds several hundreds of terabytes when 55 intra-pigment modes are considered per site (see Supplementary Note 4) and, therefore, is infeasible with current computer architectures. In this work, we employ T-TEDOPA method where an experimentally estimated vibrational spectral density is mapped to a one-dimensional chain of quantum harmonic oscillators whose complexity is unaffected by the number of long-lived intra-pigment modes in the spectral density. We also employ optimized HEOM method where simulation parameters are determined by fitting the bath correlation function of highly structured environments for a finite time window corresponding to the line width of experimentally measured absorption spectra. These two methods enable one to consider the full environmental structures of WSCP and SP with a moderate simulation cost of the order of a few gigabytes or less (see Supplementary Notes 3 and 4). In addition, numerically exact results obtained by these two independent methods coincide, demonstrating the high accuracy and reliability of our simulated data (see Supplementary Note 6).

**WSCP homodimer.** The electronic parameters of PPCs have been estimated based on a comparison of experimentally measured spectroscopic data with approximate theoretical results where environmental structures are coarse-grained or vibronic couplings are treated perturbatively. Based on a coarse-grained spectral density $J_l^{\mathrm{B777}}(\omega)$, shown in red in Fig. 2a, a best fit to the experimental absorption spectra of WSCP homodimers implies

an electronic coupling strength estimate of $V \approx 70\,\text{cm}^{-1}$ [61], as shown in red in Fig. 2b. Such an electronic coupling results in an excitonic splitting $\Delta \approx 2V \approx 140\,\text{cm}^{-1}$ which is consistent with the experimentally observed energy-gap between two absorption peaks at 656 and 662 nm, respectively. Since all the high frequency intra-pigment modes are neglected in the coarse-grained spectral density and the energy-gap between absorption peaks is smaller than the vibrational frequencies of the intra-pigment modes ($\Delta < \omega_k$), the estimated value could be interpreted as the effective coupling $V_{00}$ between $|\varepsilon_1,0\rangle$ and $|\varepsilon_2,0\rangle$ where $|0\rangle$ denotes the common vibrational ground state of the intra-pigment modes in the electronic excited state manifold. As shown in Fig. 3a, the transition dipole strength between $|g,0\rangle$ and $|\varepsilon_i,0\rangle$ (0-0 transition) of a monomer is reduced by a factor of $\exp(-\sum_k s_k/2)$, as the total transition dipole strength of the monomer is redistributed to 0-1 transitions between $|g,0\rangle$ and $|\varepsilon_i,1_k\rangle$ where only the $k$-th mode is singly excited (see Supplementary Note 10). As a result, the effective coupling between 0-0 transitions, shown in Fig. 3b, is reduced to $V_{00} = V \exp(-\sum_k s_k)$ depending on the HR factors $s_k$ of the intra-pigment modes. This implies that $V_{00} \approx 70\,\text{cm}^{-1}$ corresponds to a bare electronic coupling $V = V_{00} \exp(\sum_{k=1}^{55} s_k) \approx 2V_{00} \approx 140\,\text{cm}^{-1}$ under the full environmental spectral density $J_l^{\text{WSCP}}(\omega) + J_h(\omega)$, including the 55 intra-pigment modes shown in black in Fig. 2a. The renormalised electronic coupling $V \approx 140\,\text{cm}^{-1}$ yields a best fit to experimentally measured absorption spectra, as shown in black in Fig. 2c, when all the $M = 55$ intra-pigment modes are considered in simulations. The energy-gap between absorption peaks is gradually reduced from excitonic splitting $\Delta \approx 2V \approx 280\,\text{cm}^{-1}$ to $\Delta' \approx 2V_{00} \approx 140\,\text{cm}^{-1}$, as the number $M$ of the lowest-frequency intra-pigment modes considered in simulations is increased from 20 via 40 to 55 (see Fig. 2a, c). The electronic coupling $V \approx 70\,\text{cm}^{-1}$ estimated based on the coarse-grained low-frequency spectral density cannot reproduce the experimental results when the full spectral density is considered in simulations, as shown in Fig. 2d. The energy-gap between absorption peaks shown in Fig. 2c, d can be quantitatively well described by the splitting of 0-0 transitions, $2V_{00} = 2V \exp(-\sum_{k=1}^{M} s_k)$, implying that the effective couplings $V_{01}$ between 0-0 and 0-1 transitions, schematically shown in Fig. 3b, are not strong enough to modify the energy-gap between low-energy absorption peaks of WSCP. However, the weak $V_{01}$ couplings can redistribute the transition dipole strength from 0-0 to 0-1 transitions and significantly modify the high-energy part of absorption spectra, which cannot be described by conventional line shape theory (see Supplementary Note 10).

**Multi-mode vibronic mixing in exciton basis.** In contrast to WSCP, the bare excitonic splitting of SP is of the order of the typical vibrational frequencies of the intra-pigment modes and the resulting redistribution of oscillator strengths and shifts of optical lines are much more difficult to predict. To qualitatively estimate these effects, we consider second-order perturbation theory starting from the full Hamiltonian $H = H_e + H_\nu + H_{e-\nu}$ in the single-exciton manifold. In that case, the vibronic mixing is induced by the relative motion of the intra-pigment modes with identical frequency $\omega_k$, described by $b_k = (b_{1,k} - b_{2,k})/\sqrt{2}$, as the center of mass motion, described by $B_k = (b_{1,k} + b_{2,k})/\sqrt{2}$, merely induces the homogeneous broadening of absorption line shapes without affecting exciton dynamics (see Supplementary Note 1). Hence, we can discard the center-of-mass part of the total Hamiltonian to find $H = H_0 + H_I$ where

$$H_0 = H_e + H_\nu + \cos(\theta)\,\sigma_z \sum_{k=1}^{55} \omega_k \sqrt{s_k/2}(b_k + b_k^\dagger), \quad (5)$$

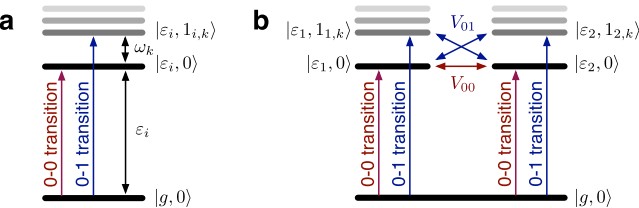

**Fig. 3 Vibronic energy-levels in site basis. a** Energy-level structure of monomer with 0-0 and 0-1 transitions highlighted in red and blue, respectively. **b** Energy-level structure of dimer with $V_{00}$ and $V_{01}$ representing the effective coupling between 0-0 transitions and the interaction between 0-0 and 0-1 transitions, respectively.

with $H_\nu = \sum_k \omega_k b_k^\dagger b_k$, and

$$H_I = -\sin(\theta)\,\sigma_x \sum_{k=1}^{55} \omega_k \sqrt{s_k/2}(b_k + b_k^\dagger). \quad (6)$$

Here $\theta = \tan^{-1}[2V/(\varepsilon_1 - \varepsilon_2)]$, while $\sigma_x = |E_+\rangle\langle E_-| + |E_-\rangle\langle E_+|$ and $\sigma_z = |E_+\rangle\langle E_+| - |E_-\rangle\langle E_-|$ are the Pauli matrices in the exciton basis. The Hamiltonian $H_0$ is diagonalised by the polaron transformation in the exciton basis, $U = |E_+\rangle\langle E_+|D_\theta + |E_-\rangle\langle E_-|D_\theta^\dagger$ with $D_\theta = \exp[\cos(\theta)\sum_k \sqrt{s_k/2}(b_k^\dagger - b_k)]$. For typical HR factors of PPCs, of the order of $s_k \lesssim 0.01$, the vibronic mixing is dominated by contributions from the single vibrational excitation subspace where it leads to eigenstates of $H$ of the form

$$|\psi_\pm\rangle = a_{\pm,0}|E_\pm,0\rangle + \sum_{k=1}^{55} a_{\mp,1_k}|E_\mp,1_k\rangle, \quad (7)$$

with $|0\rangle$ and $|1_k\rangle$ representing vibrational states where all the intra-pigment modes are in their ground states or only one mode described by $b_k$ is singly excited. In second-order perturbation theory, these vibronic eigenstates $|\psi_\pm\rangle$ have energies

$$E_\pm' = E_\pm \pm \alpha \frac{2V^2}{\Delta^2} \sum_{k=1}^{55} \frac{s_k \omega_k^2}{\Delta \mp \omega_k}, \quad (8)$$

and the purely excitonic splitting $\Delta = E_+ - E_-$ is shifted to a vibronic splitting

$$\Delta' = E_+' - E_-' = \Delta\left(1 + \alpha \frac{4V^2}{\Delta^2} \sum_{k=1}^{55} \frac{s_k \omega_k^2}{\Delta^2 - \omega_k^2}\right), \quad (9)$$

where $\alpha = \exp(-2\cos^2(\theta) \sum_{k=1}^{55} s_k)$. These energetic corrections are in complete analogy to the well-known light shifts in atomic physics. The sign of these energy shifts is determined by the difference in excitonic splitting and vibrational frequency, $\Delta - \omega_k$. We note that the vibronic energy renormalization can also be described in the regular electronic-vibrational basis without the polaron transformation using second order perturbation theory (see Supplementary Note 2).

For an excitonic splitting that is smaller than the vibrational frequencies, $\Delta \lesssim \omega_k$, the energy-gap $\Delta'$ between vibronic eigenstates $|\psi_+\rangle$ and $|\psi_-\rangle$ is reduced compared to the bare excitonic splitting $\Delta$ (see Fig. 4a). This is in line with our numerically exact simulations of WSCP where the bare excitonic splitting $\Delta \approx 2V$ is reduced to $\Delta' \approx 2V_{00} \approx V$. It is notable that for PPCs consisting of chlorophylls or bacteriochlorophylls, the HR factors of the intra-pigment modes are of the order of $s_k \approx 0.01$, independent of the vibrational frequencies $\omega_k$. In case the excitonic splitting is significantly smaller than the vibrational frequencies of the intra-pigment modes, the detuning between them is well approximated by $\Delta_k = \omega_k - \Delta \approx \omega_k$, thus exhibiting the same scaling in $\omega_k$ as the electronic-vibrational coupling, $g_k = \omega_k \sqrt{s_k}$. This implies that the coupling of higher-frequency modes increases with the detuning

$\Delta_k$ so that they cannot simply be ignored on the basis of being off-resonant.

When the excitonic splitting is larger than the vibrational frequencies, $\Delta \gtrsim \omega_k$, the situation is reversed (see Fig. 4b), resulting in an increased vibronic splitting $\Delta'$ compared to the bare excitonic splitting $\Delta$. This case cannot be described by the splitting of 0-0 transitions, since the effective coupling $V_{00} = V \exp(-\sum_k s_k)$ is smaller in magnitude than a bare electronic coupling $V$ for arbitrary HR factors defined by $s_k \geq 0$. This implies that the mixing of 0-0 and 0-1 transitions can result in two absorption peaks with an energy gap $\Delta'$ being larger than the bare excitonic splitting $\Delta$.

**Special pair in bacterial reaction center.** The photosynthetic reaction center which drives exciton dissociation into free charges consists of the SP and four additional pigments[63]. The SP is a strongly coupled dimeric unit with an electronic coupling estimated to be $V = 625 \, \mathrm{cm}^{-1}$, a difference in mean site energies of $\langle \varepsilon_1 - \varepsilon_2 \rangle = 315 \, \mathrm{cm}^{-1}$ and consequently a bare excitonic splitting of $\Delta \approx 1290 \, \mathrm{cm}^{-1}$. These electronic parameters have been

estimated based on a best fit to absorption, linear dichroism, and hole burning spectra of bacterial reaction centers using conventional line shape theory[51]. In what follows, we neglect the order of magnitude weaker electronic coupling of the SP to the four additional pigments and do not aim to reproduce experimentally measured absorption spectra of the whole bacterial reaction centers and re-estimate electronic parameters. Rather we concentrate on the effect of multi-mode vibronic mixing on the SP and its consequences regarding the nature and lifetimes of excitonic coherence and long-lived oscillatory signals in 2D electronic spectra.

While in WSCP the excitonic splitting is far detuned from high-frequency modes, the situation is markedly different for the SP. Here the environmental spectral density contains high-frequency intra-pigment modes both above and below the bare excitonic gap, as shown in black in Fig. 5a. The smaller frequency differences between vibrational modes and excitonic splitting and the varying sign of their detuning makes the effect of multimode mixing harder to predict analytically. Indeed, the perturbation procedure for obtaining Eq. (9) will be inaccurate for a larger number of modes. The vibronic splitting can be estimated beyond the perturbation theory by numerically diagonalising the Hamiltonian $H = H_0 + H_I$ in Eqs. (5), (6), leading to $\Delta' \approx 1744 \, \mathrm{cm}^{-1}$ (see Supplementary Note 7). This estimate is in line with numerically exact simulated results where the energy-gap between absorption peaks is approximately $1710 \, \mathrm{cm}^{-1}$ (see 780 and 900 nm peaks in Fig. 5b, corresponding to $|\psi_+\rangle$ and $|\psi_-\rangle$, respectively) and the oscillatory dynamics of excitonic coherence is dominated by $1755 \, \mathrm{cm}^{-1}$ frequency component (see Fig. 5c). We note that the difference between excitonic and vibronic splittings is significant, of the order of $\Delta' - \Delta \approx 465 \, \mathrm{cm}^{-1}$, and this shift cannot be described by conventional line shape theory where multi-mode vibronic mixing is ignored and as a result the

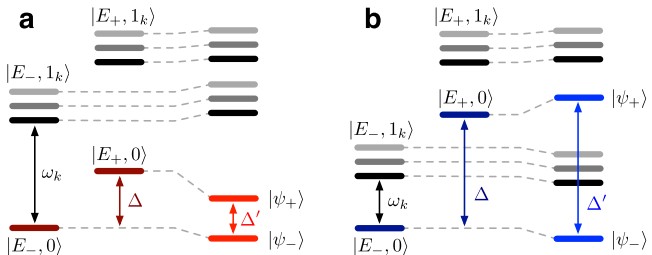

**Fig. 4 Vibronic energy-levels in exciton basis. a, b** Effect of multi-mode vibronic mixing on vibronic energy-level structure when excitonic splitting $\Delta$ is smaller (larger) than vibrational frequencies $\omega_k$ of intra-pigment modes, leading to reduction (increment) of the energy gap $\Delta'$ between vibronic eigenstates.

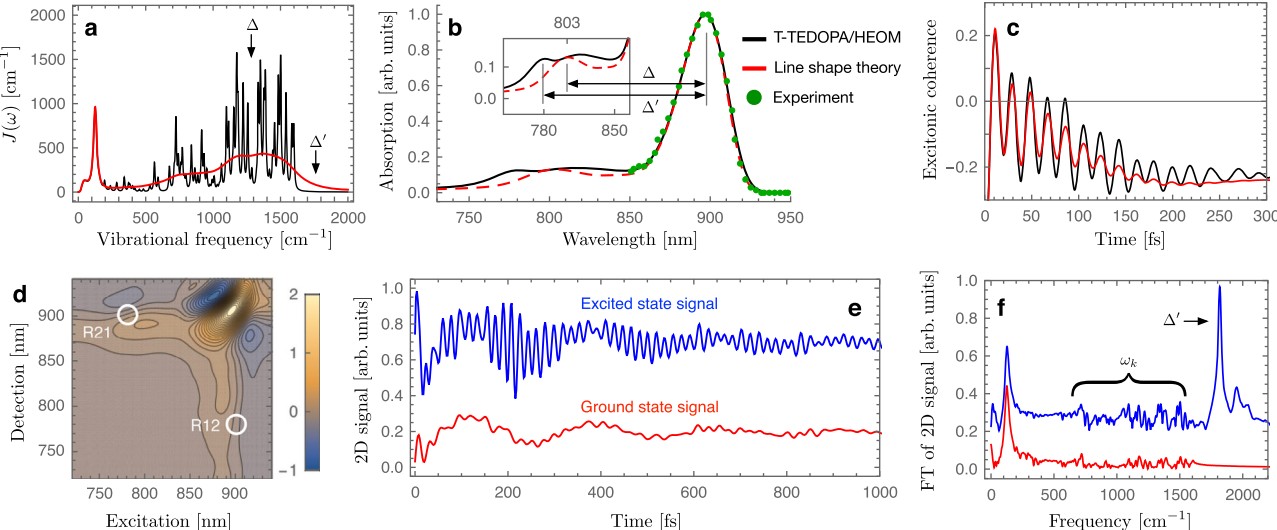

**Fig. 5 Absorption and 2D electronic spectra of SP. a** Experimentally estimated spectral density of the SP[15, 61] is shown in black for an intra-pigment mode vibrational damping rate $\gamma_k = (1 \, \mathrm{ps})^{-1}$. Coarse-grained version for $\gamma_k = (50 \, \mathrm{fs})^{-1}$ is shown in red and the excitonic and vibronic splittings, $\Delta \approx 1290 \, \mathrm{cm}^{-1}$ and $\Delta' \approx 1800 \, \mathrm{cm}^{-1}$, are highlighted. **b** Experimental absorption spectrum of the bacterial reaction center at 5 K, shown in green dots, and numerically exact absorption line shape, obtained by TEDOPA and HEOM, of the SP, shown in black. Approximate absorption spectrum of the SP computed by second-order cumulant expansion is shown in red where the energy-gap between absorption peaks at 803 and 897 nm is approximately $\Delta \approx 1300 \, \mathrm{cm}^{-1}$.
**c** Excitonic coherence dynamics for the experimentally estimated and coarse-grained environmental structures, shown in black and red, respectively, when only site 1 is initially excited. **d** Rephasing 2D spectra of the SP at waiting time $T = 0$. **e, f** 2D signals at a cross-peak R12, marked in (**d**), and corresponding Fourier transformation where ground and excited state signals are shown in red and blue, respectively. Note that excited state signals are dominated by vibronic coherence $|\psi_+\rangle\langle\psi_-|$, leading to 2D oscillations with frequency $\Delta' \approx 1800 \, \mathrm{cm}^{-1}$. The transient of the other cross-peak R21 is provided in Supplementary Note 8 and all molecular parameters used in these simulations are given in Supplementary Note 5.

energy-gap between absorption peaks is reduced to the excitonic splitting (see the inset in Fig. 5b).

**Long-lived multi-mode vibronic coherence**. The considerable size of the multi-mode mixing effects on excitonic energy gaps suggest a possibly significant influence on coherent excitonic dynamics. The coarse-grained spectral density shown in red in Fig. 5a, which corresponds to a vibrational lifetime of $\gamma_k = (50\,\text{fs})^{-1}$, yields short-lived oscillatory dynamics of excitonic coherence $\rho_{\pm}(t) = \left\langle E_{-}|\hat{\rho}_e(t)|E_{+}\right\rangle$ with $\hat{\rho}_e(t)$ denoting reduced electronic density matrix (see red line in Fig. 5c). Even if a few intra-pigment modes near-resonant with excitonic splitting are selected to be weakly damped, $\gamma_k = (1\,\text{ps})^{-1}$, the vibronic mixing with the large number of remaining strongly-damped modes, $\gamma_k = (50\,\text{fs})^{-1}$, suppresses the lifetime of excitonic coherences, making the resulting dynamics essentially identical to that where all the modes are strongly damped (see Supplementary Note 7 for detailed analysis of multi-mode vibronic mixing). In sharp contrast, when the picosecond lifetime of actual intra-pigment modes is considered, $\gamma_k = (1\,\text{ps})^{-1}$, the excitonic coherence dynamics is dominated by long-lived oscillations with frequency $\Delta' \approx 1755\,\text{cm}^{-1}$, associated with the vibronic coherence between $|\psi_{+}\rangle$ and $|\psi_{-}\rangle$ states (see black line in Fig. 5c).

In 2D electronic spectroscopy, the third-order nonlinear optical response of molecular systems is measured by using a sequence of femtosecond pulses with controlled time delays[64,65]. As is the case of pump probe experiments[66], electronically excited state populations and coherences can be created by a pair of pump pulses, and the molecular dynamics in the electronic excited state manifold can be monitored by controlling the time delay $T$ between pump and probe. The additional time delay between two pump pulses enables one to monitor the molecular dynamics as a function of excitation and detection wavelengths for each waiting time $T$. The optical transitions induced by the pump pulses can also create vibrational coherences in the electronic ground state manifold, making it challenging to extract the information about coherent electronic dynamics from multidimensional spectroscopic data[46].

Our numerically exact simulations of the SP demonstrate that long-lived oscillatory signals in 2D electronic spectra can originate from purely vibrational coherences or from vibronic coherences induced by multi-mode mixing. The latter have been ignored in previous numerical studies which considered only a few intra-pigment modes quasi-resonant with excitonic splitting and neglected all the modes that are far detuned from excitonic transitions as they were deemed to have a negligible effect[67]. However, the correct assessment of the nature of oscillatory 2D signals requires the computation of 2D spectra under the influence of the full spectral density. In order to make such computation feasible, in Supplementary Note 8, we provide an approximate master equation for vibronic dynamics, which takes into account multi-mode mixing effects and quantitatively reproduces numerically exact absorption line shape of the SP. Figure 5d shows the resulting rephasing 2D spectra at waiting time $T = 0$ in the presence of inhomogeneous broadening. The 2D lineshape, shown as a function of excitation and detection wavelengths, is dominated by a diagonal peak excited and detected at 900 nm which coincides with the position of the main absorption peak (see Fig. 5b). To investigate the excited state coherence between vibronic eigenstates $|\psi_{+}\rangle$ and $|\psi_{-}\rangle$, which induce the absorption peaks at 780 and 900 nm, respectively, we focus on a cross-peak R12 marked in Fig. 5d. Figure 5e shows the transient of the cross-peak as a function of the waiting time $T$ where the oscillatory 2D signals originating from electronic ground state manifold, shown in red, are comparable to those of excited state signals, shown in blue. The ground state signals

consist of multiple frequency components below $1600\,\text{cm}^{-1}$, corresponding to the vibrational frequencies $\omega_k$ of underdamped intra-pigment modes, as shown in Fig. 5f. It is important to note that the excited state signals include a long-lived oscillatory component with frequency ~$1800\,\text{cm}^{-1}$, which is not present in the ground state signals and cannot originate from purely vibrational effects as they exceed the high-frequency cut-off of the environmental spectral density (see Fig. 5a). This component must therefore originate from long-lived vibronic coherence due to multi-mode mixing. The long-lived oscillations at $\Delta' \approx 1800\,\text{cm}^{-1}$ frequency cannot be described by coarse-grained environment models where only a few intra-pigment modes near-resonant with the excitonic splitting $\Delta \approx 1300\,\text{cm}^{-1}$ are weakly damped ($\gamma_k = (1\,\text{ps})^{-1}$), while all the other intra-pigment modes are strongly damped ($\gamma_k = (50\,\text{fs})^{-1}$) or neglected ($s_k = 0$) in 2D simulations (see Supplementary Note 8). Our results demonstrate that while some oscillatory components in 2D spectra can originate from purely vibrational motions, long-lived 2D oscillations can also be the result of a strong vibronic mixing of excitons with a large number of underdamped intra-pigment modes.

## Discussion

Employing numerically exact methods and an analytical theory, we have investigated exciton-vibrational dynamics under the complete vibrational spectrum that has been estimated in earlier experiments. We considered two paradigmatic regimes. The first regime, represented by an excitonic dimer in WSCP, is characterized by an excitonic splitting that is smaller than vibrational frequencies of intra-pigment modes. In this case, one main effect of vibronic coupling to the intra-pigment modes is a reduction of the dipole strength of 0-0 transitions of monomers and of their effective coupling strength $V_{00}$ that determines the splitting between absorption peaks in the low-energy spectrum. A second important effect concerns the modulation of the vibrational sideband of optical transitions by a vibronic mixing between 0-0 and 0-1 transitions. Although the vibronic mixing is not strong enough to modulate the low-energy part of absorption spectra of WSCP, it can induce a notable dipole strength redistribution between 0-0 and 0-1 transitions, which cannot be described by approximate theories where the vibronic mixing is ignored.

In the second regime, represented by the SP of the photosynthetic reaction center of purple bacteria, the excitonic splitting is located in the middle of the high frequency part of the intra-pigment vibrational spectrum. In this case, the splitting between main absorption peaks can be even larger than the bare excitonic splitting, due to multi-mode vibronic mixing effects. This regime is found to be particularly suitable for the discovery of new long-lived quantum coherences in photosynthesis. We found that the coherence time of excitonic dynamics is not simply governed by the lifetime of quasi-resonant intra-pigment modes. Rather it is determined by the lifetimes of individual intra-pigment modes involved in a multi-mode vibronic mixing. This implies that approximate theoretical models based on coarse-graining of the high frequency part of the vibrational environments[21] may underestimate the lifetime of excitonic coherences and could be inappropriate to analyze quantum coherences observed in nonlinear experiments on photosynthetic systems. In addition, our results demonstrate that even if the frequency $\Delta'$ of oscillatory 2D signals is not well matched to one of the vibrational frequencies $\omega_k$ of intra-pigment modes, the long-lived 2D oscillations can be vibronic in origin, rather than being purely electronic, as is the case of the SP where $\omega_k \lesssim 1600\,\text{cm}^{-1} < \Delta' \approx 1800\,\text{cm}^{-1}$. This implies that the origin of long-lived oscillatory 2D signals cannot be identified based only on a comparison of the frequency spectrum of nonlinear signals with the vibrational frequency spectrum of underdamped modes. Hence,

we contend that previously ignored multi-mode vibronic effects must be included in the interpretation of nonlinear spectroscopic signals before the current debate regarding the presence and nature of long-lived quantum coherences in pigment-protein complexes can be settled conclusively.

Our results suggest the possibility that the energy transfer dynamics between electronic states, such as excitons and charge-transfer states, could be governed by the multi-mode nature of the total vibrational environments, rather than a few vibrational modes quasi-resonant with electronic energy-gaps (see Supplementary Note 9). The generality of the methods employed here also suggests that our results have a broad scope and can be of relevance in a wide variety of scenarios involving strong hybridization of electronic and vibrational degrees of freedom, such as recent observations of nonadiabatic dynamics in cavity polaritonics[68,69]. We expect that renormalization effects considered here may open an entirely new toolbox for vibrational reservoir engineering with possible applications in information technologies and polaritonic chemistry.

## Data availability

The simulated absorption and 2DES data generated in this study are provided in the Source Data file. The data used in this paper are also available from the authors upon request. Source data are provided with this paper.

## Code availability

The codes used in this work are available from the authors upon reasonable request.

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

## Acknowledgements

F.C.-S., A.M., J.L., S.F.H., and M.B.P. acknowledge financial support by the ERC Synergy grants BioQ and HyperQ, and support by the state of Baden-Württemberg through bwHPC and the German Research Foundation (DFG) through grant no INST 40/575-1 FUGG (JUSTUS 2 cluster). A.M. acknowledges financial support by an IQST PhD fellowship. T.R. acknowledges financial support by the Austrian Science Fund (FWF): P 33155-NBL.

## Author contributions

F.C.-S., T.R., S.F.H., and M.B.P. initiated this work. F.C.-S., A.M., and J.L. performed numerical simulations. All authors discussed the results and contributed to the writing of the paper.

## Funding

## Competing interests

The authors declare no competing interests.
