## [Peer Review File · Nature Communications]

Reviewer #1 (Remarks to the Author):

This paper handles the issue of vibronic mixing in pigment protein complexes (PPCs). By taking two exemplary PPCs, one with relatively weak coupling and small exciton splitting and the other with strong coupling and large splitting, the authors provide how things can show up differently with the vibronic mixing. Technically, they adopt polaron bases, which should be ideal for handling vibronic mixing as they can treat electron and phonon degrees of freedom on an equal footing. Thus, combining polaron basis and numerically exact method like HEOM or TEDOPA will be a charming approach, and I even wonder why no one really tried it before like here toward simulating 2DES features. Through the approaches, the authors show that many experimental features can be explained with vibronic mixing that involves multiple modes (not just the exciton gap matching mode or low frequency phonon band).

I think the paper can benefit the readership of Nat Commun in general and am fully supportive of its publication, but I wish the authors to consider the following aspects before the actual publication.

1. Adjusting the overall structure:

I feel the Results section presently is presenting topics in too much scattered way. I can see why the authors put things that way, but it is still distracting. One easy way of seeing what I feel is to see only the subsection titles. Perhaps, the authors need to boldly prune some stuffs out and graft them into a new section after Discussion?

2. Physical nature of multimode mixing:

The authors contend that "previously ignored multimode vibronic effects must be included in the interpretation of nonlinear spectroscopic signals before the current debate regarding the presence and nature of long-lived quantum coherences in PPCs can be settled conclusively". Can they provide more on the physical interpretation of this multimode effect? Is it related to vibration-vibration mixing? If one use regular electronic / vibrational basis (not polaron basis), is it something that will show up as high order perturbations?

3. In a nutshell, we have multiple e's and multiple v's, and there are e-e, e-v, and v-v mixing. By construction, the v-v mixing is not explicitly treated with the harmonic oscillator model, but can still be there from (1) indirect coupling through v_i -e (+) e- v_j couplings and (2) damping effects related to γ_k . Does the fact that the multimode effect is important mean that one should consider the v-v mixing importantly?

4. In my experience, vibronic resonance enhancement of an energy transfer rate is a very broad feature. Namely, an energy transfer between two electronic states can be readily mediated by a vibration as long as its frequency is close enough, but not perfectly matching with the electronic gap. The energy discrepancy between the electronic gap and the vibrational frequency means that some additional vibrations should get involved for fulfilling the law of energy conservation. This signifies that multimode mixing is somehow kicking in there too in energy transfers. Accounting on this aspect may be an interesting discussion.

Reviewer #2 (Remarks to the Author):

The authors investigate effect of the high-frequency part of intra-pigment vibrations in PPCs based on a widely-used model of exciton-vibration Hamiltonian. Their primary approach is based on an analytical method of polaron transformation and numerically-exact quantum-dynamical methods for comparison with previously obtained experimental data. By performing simulations, the authors demonstrate the following three phenomena caused by multi-mode vibrational mixing, which have been overlooked in previous studies:

- 1) splitting between absorption peaks in the experimentally obtained spectrum of WSCP,
- 2) possible deviation from conventional line shape theory in absorption spectrum of SP,
- 3) possible long-lived coherence in excited state signals of 2D electronic spectroscopy.

The authors then claim that the multi-mode vibrational mixing is not ignorable and hence must be considered in analysis of nonlinear experiments on the PPCs.

I find their result on 1) particularly convincing, and their absorption spectrum reproduced by simulation is clearly more accurate than that obtained based on the coarse grained approach on the high-frequency vibrations which has been commonly employed in analysis of the PPCs. On 2) and 3), their simulations are restricted to the exciton dynamics of the SP, which is only a part of the reaction

center complex, making them difficult to be justified experimentally. However, their claim is reasonable if there are still unexplained portion of coherences existing in 2D experiments.

A couple of suggestions to the authors:

- On Fig. 5f, blue line is showing the peak around 1800 cm^{-1} that the authors find due to the multi-mode mixing. It would be helpful for a reader if there is a graphical comparison between the multi-mode mixing and the course-grained approaches to see the effect quantitatively at least in that frequency range.

- The authors present the coherences extracted from the reduced density matrix of their simulations on the SP, but no discussions are found on population dynamics. A largely debated topic in the community of photosynthesis is on possible biological functions of the coherences in energy and charge transport in PPCs. Since the physical significance of the multi-mode mixing is emphasized here, it is natural to raise a question as to whether the mixing enhances exciton transport or not, especially within the SP that is considered to work as the active site of charge separation. Therefore, it would be helpful for a reader if the authors could comment on their population analysis.

Also, in line 506, I believe "that" is to be "than."

Reply to Reviewer #1

This paper handles the issue of vibronic mixing in pigment protein complexes (PPCs). By taking two exemplary PPCs, one with relatively weak coupling and small exciton splitting and the other with strong coupling and large splitting, the authors provide how things can show up differently with the vibronic mixing. Technically, they adopt polaron bases, which should be ideal for handling vibronic mixing as they can treat electron and phonon degrees of freedom on an equal footing. Thus, combining polaron basis and numerically exact method like HEOM or TEDOPA will be a charming approach, and I even wonder why no one really tried it before like here toward simulating 2DES features. Through the approaches, the authors show that many experimental features can be explained with vibronic mixing that involves multiple modes (not just the exciton gap matching mode or low frequency phonon band).

I think the paper can benefit the readership of Nat Commun in general and am fully supportive of its publication, but I wish the authors to consider the following aspects before the actual publication.

Comment 1. *Adjusting the overall structure:*

I feel the Results section presently is presenting topics in too much scattered way. I can see why the authors put things that way, but it is still distracting. One easy way of seeing what I feel is to see only the subsection titles. Perhaps, the authors need to boldly prune some stuffs out and graft them into a new section after Discussion?

The authors: We appreciate the advice of the referee and have reassessed carefully the structure of our manuscript. While most of the material is, in our opinion, essential to make our scientific case, we have come to realise that the subsection entitled “Multi-mode vibronic effects in the site basis”, which provides the mathematical description of the transition dipole moment redistribution within a monomer and an effective coupling between 0-0 transitions of a dimer, does not have to be included in the main text and can be, as the referee noticed, somewhat distracting. We have therefore decided to move this description to Supplementary Note 10 and added a brief explanation of these concepts in the Main Text (see item 1 in the list of changes). We believe that this restructuring will improve the readability of our manuscript.

Comment 2. *Physical nature of multimode mixing:*

The authors contend that "previously ignored multimode vibronic effects must be included in the interpretation of nonlinear spectroscopic signals before the current debate regarding the presence and nature of long-lived quantum coherences in PPCs can be settled conclusively". Can they provide more on the physical interpretation of this multimode effect?

The authors: Typically, the electronic parameters of PPCs are estimated on the basis of fitting of linear optical spectra based on vibronic models where excitons are only coupled to intra-pigment modes whose vibrational frequencies are close to the excitonic splitting. However, our work based on numerically exact calculations shows that the estimated values of the electronic parameters of

PPCs can change significantly when multi-mode vibronic effects are accounted for. Specifically, as we show, the high-frequency modes that are typically neglected owing to their large detuning couple to excitons with a strength that is proportional to their frequency so that independently of the detuning they lead to significant vibronic energy shifts in the low energy manifold (see e.g. the discussion around eqs. 5-9). These shifts affect energetic resonances between excitonic and vibrational states whose explanation, without accounting for the presence of the high energy modes, requires significantly modified electronic coupling strengths.

As a result, when the electronic parameters are re-estimated by taking into account the multi-mode effects, the theoretical predictions of energy transfer dynamics and associated nonlinear optical responses of PPCs can change quantitatively to an extent that change the way the long-lived oscillatory features in nonlinear optical spectra have been interpreted in literature.

More specifically, our work demonstrates that a strong vibronic mixing can be present even if the frequency of oscillatory nonlinear signals is not well matched to the vibrational frequencies of the intra-pigment modes. For the case of the special pair (SP), we demonstrated that an excitonic splitting of 1300 cm^{-1} can be shifted to a vibronic splitting of 1800 cm^{-1} , when exciton states are vibronically mixed with 55 intra-pigment modes. The frequency of oscillatory nonlinear signals, 1800 cm^{-1} , observed in our simulations is beyond the highest vibrational frequency of the intra-pigment modes, 1600 cm^{-1} , considered in our simulations. Therefore, when conventional approaches are used to interpret the origin of the oscillatory signals at 1800 cm^{-1} frequency, one would mistakenly infer that an excitonic splitting is close to the observed frequency 1800 cm^{-1} and the long-lived coherences are electronic in origin.

To emphasize the importance of multimode vibronic effects in the interpretation of nonlinear spectra, we have added the following sentences to the Discussion section in the Main Text (see item 2 in the list of changes): “... *In addition, our results demonstrate that even if the frequency Δ' of oscillatory 2D signals is not well matched to one of the vibrational frequencies ω_k of intra-pigment modes, the long-lived 2D oscillations can be vibronic in origin, rather than being purely electronic, as is the case of the SP where $\omega_k \leq 1600\text{ cm}^{-1} < \Delta' \approx 1800\text{ cm}^{-1}$. This implies that the origin of long-lived oscillatory 2D signals cannot be identified based only on a comparison of the frequency spectrum of nonlinear signals with the vibrational frequency spectrum of underdamped modes.*”

Comment 2 continued: *Is it related to vibration-vibration mixing?*

The authors: The multi-mode vibronic mixing is mainly described by a vibronic eigenstate of the form $|\psi_+\rangle = A_0|E_+, 0\rangle + \sum_k A_k|E_-, 1_k\rangle$ where $|E_+\rangle$ and $|E_-\rangle$ denote higher- and lower-energy exciton states, respectively, $|0\rangle$ the state where all intra-pigment modes are in their ground states, and $|1_k\rangle$ a single vibrational excitation in the k -th mode only while all the other intra-pigment modes are in their ground states. The coherence between vibronic eigenstates $|\psi_-\rangle = |E_-, 0\rangle$ and $|\psi_+\rangle$ is expressed as $|\psi_+\rangle\langle\psi_-| = A_0|E_+, 0\rangle\langle E_-, 0| + \sum_k A_k|E_-, 1_k\rangle\langle E_-, 0|$

where the first term describes an excitonic coherence component, while the latter terms include the vibrational coherences of multiple intra-pigment modes.

The vibration-vibration mixing may occur in a vibrational sideband of a lower-energy exciton. When the $|E_-, 1_k\rangle$ state interacts with $|E_+, 0\rangle$ state and then with $|E_-, 1_{k'}\rangle$, through an indirect coupling, the $|E_-, 1_k\rangle$ state, where the k -th intra-pigment mode is singly excited, can be mixed with the other $|E_-, 1_{k'}\rangle$ states, where a different single mode k' is excited instead of the mode k . As shown in Supplementary Figure 3a and b, and Supplementary Note 7, the vibronic eigenstates contributing to a vibrational sideband can have a form $|\psi_k\rangle = \sum_k B_k |E_-, 1_k\rangle + B_0 |E_+, 0\rangle$ with $|B_0| \ll 1$. However, the indirect coupling is not strong enough to induce any notable energy-level shift between $|E_-, 1_k\rangle$ and $|\psi_k\rangle$. This is in contrast to the multi-mode vibronic mixing described by $|\psi_+\rangle$ where $|E_+, 0\rangle$ is directly coupled to multiple $|E_-, 1_k\rangle$ states, which can induce a large energy-level shift between $|E_+, 0\rangle$ and $|\psi_+\rangle$.

Comment 2 continued: *If one use regular electronic / vibrational basis (not polaron basis), is it something that will show up as high order perturbations?*

The authors: The multi-mode vibronic effects show up as second order perturbations in both regular and polaron bases. The vibronic splitting computed by second order perturbation theory in the regular basis has been summarized in Supplementary Note 2, which has been briefly mentioned in the Main Text (see item 3 in the list of changes).

Comment 3. *In a nutshell, we have multiple e's and multiple v's, and there are e-e, e-v, and v-v mixing. By construction, the v-v mixing is not explicitly treated with the harmonic oscillator model, but can still be there from (1) indirect coupling through v_i-e (+) e-v_j couplings and (2) damping effects related to gamma_k. Does the fact that the multimode effect is important mean that one should consider the v-v mixing importantly?*

The authors: As explained in the response to Comment 2 above, the indirect coupling between $|E_-, 1_k\rangle$ and $|E_-, 1_{k'}\rangle$ mediated by $|E_+, 0\rangle$ can induce a vibration-vibration mixing. However, it does not induce any notable energy-level shifts of the vibronic eigenstates $|\psi_k\rangle$ contributing to a vibrational sideband due to the weak indirect coupling strength and therefore it is not important for the model parameters considered in our work, although there could be situations where this phenomenon may become relevant.

Comment 4. *In my experience, vibronic resonance enhancement of an energy transfer rate is a very broad feature. Namely, an energy transfer between two electronic states can be readily mediated by a vibration as long as its frequency is close enough, but not perfectly matching with the electronic gap. The energy discrepancy between the electronic gap and the vibrational frequency means that some additional vibrations should get involved for fulfilling the law of energy conservation. This signifies that multimode mixing is somehow kicking in there too in energy transfers. Accounting on this aspect may be an interesting discussion.*

The authors: We agree with the referee that a discussion on energy transfer in the presence of all 55 intrapigment modes is pertinent here. For simplicity, we consider the case where the intra-pigment modes are undamped and disregard initially the low-frequency continuous phonon spectrum. Later on in our discussion we will add this low-frequency spectrum again. Starting from an initial state $|E_+, 0\rangle$ with $|E_+\rangle$ and $|0\rangle$ denoting, respectively, a higher-energy exciton state and a global vibrational ground state where all the intra-pigment modes are in their vacuum states, the energy exchange between excitons and vibrational modes induces transitions to lower-energy exciton states $|E_-, 1_k\rangle$, where only the k -th intra-pigment mode is singly excited, and $|E_-, 2_k\rangle$ and $|E_-, 1_k, 1_{k'}\rangle$, where only the k -th mode is doubly excited or two different modes k and k' are singly excited at the same time.

Fig. 1. Population transfer dynamics and vibronic resonance condition

In Fig.1a above, the population of the vibrationally cold $|E_-, 0\rangle$ state is shown in blue, while the total population of singly excited vibrational states $|E_-, 1_k\rangle$ and that of doubly excited vibrational states $|E_-, 2_k\rangle$ and $|E_-, 1_k, 1_{k'}\rangle$ are shown, respectively, in green and red. We observe that the coherent vibronic energy transfer is dominated by the singly excited vibrational states, shown in green, but that the contribution of the doubly excited vibrational states, shown in red, is not negligible. To identify how these transitions are related to the vibrational frequencies of the intra-pigment modes and the excitonic splitting of the SP, in Fig.1b above, the populations of the $|E_-, 1_k\rangle$ states at time 1 ps are shown in green dots as a function of the detuning $\omega_k - \Delta$ with ω_k denoting the vibrational frequency of the k -th intra-pigment mode, and Δ the excitonic splitting. We note that the transitions to the singly excited vibrational states $|E_-, 1_k\rangle$ occur for a wide range of vibrational frequencies ω_k due to a moderate vibronic coupling strength being comparable to the detuning $\omega_k - \Delta$. In Fig.1b above, the populations of doubly excited vibrational states $|E_-, 2_k\rangle$ and $|E_-, 1_k, 1_{k'}\rangle$ are shown in red dots as a function of $\omega_k + \omega_{k'} - \Delta$. We obtain that the transitions to the doubly excited vibrational states mainly occur when the sum of vibrational energy quanta, $\omega_k + \omega_{k'}$, is close to the vibronic splitting renormalized by multi-mode vibronic mixing, namely $\Delta' \approx \Delta + 500$ cm⁻¹, instead of the bare excitonic splitting Δ . This implies that the two-phonon transitions can contribute to the

energy transfer dynamics, as pointed out by the reviewer, but the resonance condition depends on the vibronic splitting Δ' that is renormalized by a multi-mode vibronic mixing, instead of the excitonic splitting Δ . This is similar to the transitions to the singly excited vibrational states shown in green dots where the populations of the $|E_-, 1_k\rangle$ states become higher as the vibrational energy quanta, ω_k , is closer to the vibronic splitting $\Delta' \approx \Delta + 500 \text{ cm}^{-1}$, instead of the excitonic splitting Δ .

We note that the contribution of the two-phonon transitions to energy transfer dynamics becomes more important as the Huang-Rhys factors of the intra-pigment modes are increased, meaning that the multi-phonon processes could be relevant in other vibronic systems.

These results have been summarized in Supplementary Figure 6 and 7, and Supplementary Note 9, including additional simulations where the noise induced by low-frequency phonon environments at room temperature and vibrational damping of the intra-pigment modes is considered (see item 4 in the list of changes). The following sentence has been added to the Discussion section in the Main Text: *“Our results suggest the possibility that the energy transfer dynamics between electronic states, such as excitons and charge-transfer states, could be governed by the multi-mode nature of the total vibrational environments, rather than a few vibrational modes quasi-resonant with electronic energy gaps (see Supplementary Note 9).”*

In addition, to explain how vibrational modes whose frequencies are not close to an excitonic splitting can participate in a multi-mode vibronic mixing, the following sentences have been added to the Main Text (see item 5 in the list of changes): *“It is notable that for PPCs consisting of chlorophylls or bacteriochlorophylls, the HR factors of the intra-pigment modes are of the order of $s_k \approx 0.01$, independent of the vibrational frequencies ω_k . In case the excitonic splitting is significantly smaller than the vibrational frequencies of the intra-pigment modes, the detuning between them is well approximated by $\Delta_k = \omega_k - \Delta \approx \omega_k$, thus exhibiting the same scaling in ω_k , as the electronic-vibrational coupling, $g_k = \omega_k \sqrt{s_k}$. This implies that the coupling of higher-frequency modes increases with the detuning Δ_k so that they cannot simply be ignored on the basis of being off-resonant.”*

We thank the referee for the careful consideration of our manuscript and the very pertinent suggestions.

Reply to Reviewer #2

The authors investigate effect of the high-frequency part of intra-pigment vibrations in PPCs based on a widely-used model of exciton-vibration Hamiltonian. Their primary approach is based on an analytical method of polaron transformation and numerically-exact quantum-dynamical methods for comparison with previously obtained experimental data. By performing simulations, the authors demonstrate the following three phenomena caused by multi-mode vibrational mixing, which have been overlooked in previous studies: 1) splitting between absorption peaks in the experimentally obtained spectrum of WSCP, 2) possible deviation from conventional line shape theory in absorption spectrum of SP, 3) possible long-lived coherence in excited state signals of 2D electronic spectroscopy. The authors then claim that the multi-mode vibrational mixing is not ignorable and hence must be considered in analysis of nonlinear experiments on the PPCs.

I find their result on 1) particularly convincing, and their absorption spectrum reproduced by simulation is clearly more accurate than that obtained based on the coarse grained approach on the high-frequency vibrations which has been commonly employed in analysis of the PPCs. On 2) and 3), their simulations are restricted to the exciton dynamics of the SP, which is only a part of the reaction center complex, making them difficult to be justified experimentally. However, their claim is reasonable if there are still unexplained portion of coherences existing in 2D experiments.

A couple of suggestions to the authors:

Comment 1. *On Fig. 5f, blue line is showing the peak around 1800 cm^{-1} that the authors find due to the multi-mode mixing. It would be helpful for a reader if there is a graphical comparison between the multi-mode mixing and the course-grained approaches to see the effect quantitatively at least in that frequency range.*

The authors: To investigate how the long-lived oscillatory 2D signals at 1800 cm^{-1} frequency depend on the coarse-graining of vibrational environments, we have now considered several coarse-grained models in 2D simulations. In Fig.2a below, the experimentally estimated phonon spectral density of SP is shown in grey, while a fully coarse-grained spectral density is shown in red where all the 55 intra-pigment modes are modeled by broad Lorentzian functions with a uniform width of $(50\text{ fs})^{-1}$. The transient of a cross peak R12 in 2D electronic spectra and its frequency spectrum show that for the fully coarse-grained environments, long-lived oscillatory 2D signals do not appear in the high-frequency region, including the 1800 cm^{-1} frequency component. In Fig.2b and c, we have considered partially coarse-grained spectral densities where only three or seven intra-pigment modes near-resonant with the excitonic splitting of the SP are modeled by narrow Lorentzian functions with a width of $(1\text{ ps})^{-1}$, while all the other intra-pigment modes are modeled by the broad Lorentzian functions with the width of $(50\text{ fs})^{-1}$, as shown in green and blue. In these cases, long-lived 2D oscillations occur at the vibrational frequencies of the quasi-resonant modes, weakly damped with the rate of $(1\text{ ps})^{-1}$. However, the long-lived multi-mode

vibronic coherence at 1800 cm^{-1} frequency does not appear in 2D spectra, since the lifetime of the multi-mode vibronic coherence is determined by the overall lifetime of several tens of the intra-pigment modes involved in a multi-mode vibronic mixing. When only a few near-resonant modes are weakly damped, as considered here, the other intra-pigment modes participating in a multi-mode vibronic mixing are strongly damped and as a result the multi-mode vibronic coherence and the corresponding 1800 cm^{-1} 2D oscillations decay quickly.

Fig.2. 2D electronic spectra of coarse-grained environment models

In addition, when a large number of the off-resonant intra-pigment modes are neglected in simulations by taking the corresponding Huang-Rhys factors to be zero, the excitonic splitting $\Delta \approx 1300\text{ cm}^{-1}$ of the SP is not shifted to a multi-mode vibronic splitting $\Delta' \approx 1800\text{ cm}^{-1}$, since it requires a large number of vibrational modes involved in the vibronic mixing. Therefore, also in this case, long-lived 1800 cm^{-1} oscillations do not appear in 2D spectra.

These results have been summarized in Supplementary Note 8 and Supplementary Figure 5, and the following sentence has been added to the Main Text (see item 6 in the list of changes): *“The long-lived oscillations at $\Delta' \approx 1800\text{ cm}^{-1}$ frequency cannot be described by coarse-grained environment models where only a few intra-pigment modes near-resonant with the excitonic splitting $\Delta \approx 1300\text{ cm}^{-1}$ are weakly damped ($\gamma_k = (1\text{ps})^{-1}$), while all the other intra-pigment modes are strongly damped ($\gamma_k = (50\text{fs})^{-1}$) or neglected ($s_k = 0$) in 2D simulations (see Supplementary Note 8).”*

Comment 2. *The authors present the coherences extracted from the reduced density matrix of their simulations on the SP, but no discussions are found on population dynamics. A largely debated topic in the community of photosynthesis is*

on possible biological functions of the coherences in energy and charge transport in PPCs. Since the physical significance of the multi-mode mixing is emphasized here, it is natural to raise a question as to whether the mixing enhances exciton transport or not, especially within the SP that is considered to work as the active site of charge separation. Therefore, it would be helpful for a reader if the authors could comment on their population analysis.

Fig.3. Energy transfer under full and coarse-grained environments

The authors: As an illustration, we have now investigated energy transfer dynamics for a situation where the initial state is a higher-energy exciton state of the SP. We have considered the full vibrational environments, where the exciton states are coupled to 55 intra-pigment modes, and several coarse-grained environments where only one or three or five intra-pigment modes near-resonant with the excitonic splitting of the SP are coupled to the excitons, while all the other relatively off-resonant modes are neglected in simulations. In addition, we have considered the vibrational damping of the intra-pigment modes and vibronic couplings to low-frequency phonon environments at room temperature. In Fig.3a, the population dynamics of a lower-energy exciton state is shown. The full-environment case is shown in black line, where the lower-energy exciton state is populated rapidly within a sub-100 fs time scale. When only near-resonant modes are considered, the energy transfer dynamics becomes slower as a smaller number of the modes is considered: the results for one, three, and five near-resonant modes are shown in blue, green, and red, respectively. In Fig.3b, the population dynamics of the lower-energy exciton state is shown on a picosecond time scale. It is notable that when the full environment is considered, the lower-energy exciton state is populated rapidly within a 100-fs time scale and then saturated to a value around 0.9, as shown in black. This is in contrast to the cases where only near-resonant modes are considered, where the lower-energy exciton population is efficiently transferred back to the higher-energy exciton state, leading to large-amplitude oscillations of the population dynamics in Fig.3b, as shown in blue, green and red. Therefore, the multi-mode vibrational environments can enhance the energy transfer dynamics when compared to the cases that only quasi-resonant modes are considered. Similar multi-mode vibronic enhancement in energy transfer

dynamics has been observed for various electronic coupling strengths, including 50, 100, 300 cm^{-1} , and 625 cm^{-1} of the SP considered here.

These results have been summarized in Supplementary Note 9 and Supplementary Figure 7 (see item 4 in the list of changes). The following sentences have been added to the Discuss section of the Main Text: *“Our results suggest the possibility that the energy transfer dynamics between electronic states, such as excitons and charge-transfer states, could be governed by the multi-mode nature of the total vibrational environments, rather than a few vibrational modes quasi-resonant with electronic energy gaps (see Supplementary Note 9).”*

Comment 3. Also, in line 506, I believe “that” is to be “than.”

The authors: Indeed the typo is now corrected.

We thank the referee for the very pertinent comments which have helped us to enhance the possible physical scope of our results.

List of changes

Item 1. The mathematical description of 0-0 and 0-1 transitions and the coupling between them has been moved to Supplementary Note 10 (page 24), and these concepts have been briefly explained in the Main Text (page 3, lines 203-219).

Item 2. The importance of multi-mode vibronic effects in the interpretation of long-lived oscillations in nonlinear spectra has been explained in the Discussion section of the Main Text (page 6, lines 509-519).

Item 3. Vibronic splitting estimated by second order perturbation theory without polaron transformation has been summarized in Supplementary Note 2 (page 11) and the following sentence has been added to the Main Text (page 4, lines 289-293): *“We note that the vibronic energy renormalization can also be described in the regular electronic-vibrational basis without the polaron transformation using second order perturbation theory (see Supplementary Note 2).”*

Item 4. Population transfer dynamics simulated by a Lindblad model has been summarized in Supplementary Note 9 and Supplementary Figure 6 and 7. The following sentence has been added to the Main Text (page 6, lines 525-530): *“Our results suggest the possibility that the energy transfer dynamics between electronic states, such as excitons and charge-transfer states, could be governed by the multi-mode nature of the total vibrational environments, rather than a few vibrational modes quasi-resonant with electronic energy gaps (see Supplementary Note 9).”*

Item 5. The following sentences have been added to the Main Text (page 4, lines 300-312): *“It is notable that for PPCs consisting of chlorophylls or bacteriochlorophylls, the HR factors of the intra-pigment modes are of the order of $s_k \approx 0.01$, independent of the vibrational frequencies ω_k . In case the excitonic splitting is significantly smaller than the vibrational frequencies of the intra-pigment modes, the detuning between them is well approximated by $\Delta_k = \omega_k - \Delta \approx \omega_k$, thus exhibiting the same scaling in ω_k , as the electronic-vibrational coupling, $g_k = \omega_k \sqrt{s_k}$. This implies that the coupling of higher-frequency modes increases with the detuning Δ_k so that they cannot simply be ignored on the basis of being off-resonant.”*

Item 6. 2D electronic spectra computed based on coarse-grained phonon spectral densities have been summarized in Supplementary Note 8 and Supplementary Figure 5. The following sentence has been added to the Main Text (page 5, lines 455-462): *“The long-lived oscillations at $\Delta' \approx 1800 \text{ cm}^{-1}$ frequency cannot be described by coarse-grained environment models where only a few intra-pigment modes near-resonant with the excitonic splitting $\Delta \approx 1300 \text{ cm}^{-1}$ are weakly damped ($\gamma_k = (1\text{ps})^{-1}$), while all the other intra-pigment modes are strongly damped ($\gamma_k = (50 \text{ fs})^{-1}$) or neglected ($s_k = 0$) in 2D simulations (see Supplementary Note 8).”*

Item 7. The title of the paper has been changed to “*Exact Simulation of Pigment-Protein Complexes Unveils Vibronic Renormalization of Electronic Parameters in Ultrafast Spectroscopy*” to remove punctuation from the original title “*Exact Simulation of Pigment-Protein Complexes: Vibronic Renormalization of Electronic Parameters in Ultrafast Spectroscopy.*”

All the changes in the manuscript are highlighted in the submitted marked manuscript file (see Manuscript_marked_for_review_only.pdf).